# MANTA, an integrative database and analysis platform that relates microbiome and phenotypic data

Yi-An Chen[1]*, Jonguk Park[1], Yayoi Natsume-Kitatani[1], Hitoshi Kawashima[1],
Attayeb Mohsen[1], Koji Hosomi[2], Kumpei Tanisawa[3], Harumi Ohno[3], Kana Konishi[3],
Haruka Murakami[3], Motohiko Miyachi[3], Jun Kunisawa[2], Kenji Mizuguchi[1]*

1 Laboratory of Bioinformatics, Artificial Intelligence Center for Health and Biomedical Research, National Institutes of Biomedical Innovation, Health and Nutrition, Ibaraki, Osaka, Japan, 2 Laboratory of Vaccine Materials, Center for Vaccine and Adjuvant Research and Laboratory of Gut Environmental System, National Institutes of Biomedical Innovation, Health and Nutrition, Ibaraki, Osaka, Japan, 3 Department of Physical Activity Research, National Institutes of Biomedical Innovation, Health and Nutrition, Shinjuku, Tokyo, Japan

☯ These authors contributed equally to this work.
* chenyian@nibiohn.go.jp (YC); kenji@nibiohn.go.jp (KM)

**Data Availability Statement:** All relevant data are within the paper and its Supporting Information files.

## Abstract

With an ever-increasing interest in understanding the relationships between the microbiota and the host, more tools to map, analyze and interpret these relationships have been developed. Most of these tools, however, focus on taxonomic profiling and comparative analysis among groups, with very few analytical tools designed to correlate microbiota and the host phenotypic data. We have developed a software program for creating a web-based integrative database and analysis platform called MANTA (Microbiota And pheNoType correlation Analysis platform). In addition to storing the data, MANTA is equipped with an intuitive user interface that can be used to correlate the microbial composition with phenotypic parameters. Using a case study, we demonstrated that MANTA was able to quickly identify the significant correlations between microbial abundances and phenotypes that are supported by previous studies. Moreover, MANTA enabled the users to quick access locally stored data that can help interpret microbiota-phenotype relations. MANTA is available at https://mizuguchilab.org/manta/ for download and the source code can be found at https://github.com/chenyian-nibio/manta.

## Introduction

The genetic material of microorganisms residing within or upon the surface of the human body, especially gut microbiome, live in a mutualistic relationship with the host. These associations are key contributors to the host metabolism and are usually essential for human health. The microbiota of the intestinal tract (gut microbiota) can assist in breaking down nutrients that the host cannot digest or synthesizing vitamins that the host cannot produce. Alterations in the microbiota can lead to diseases such as obesity [1–4]. Therefore, the study of microbiota has considerable importance for public health.

The improvement of next generation sequencing technology, along with the decrease in the cost of large-scale analyses, has facilitated research on microorganism communities, for example,

**Funding:** This work was supported by the Japan Society for the Promotion of Science under grant numbers 17K07268 (K.M.), 18H02150 (J.K.), 18H02674 (J.K.), 17K09604 (J.K.), and 18K17997 (K.H.); the Japan Agency for Medical Research and Development (AMED) under grant numbers 17fk0108223h0002 (J.K.), 17ek0410032s0102 (J.K.), 17fk0108207h0002 (J.K.), 17ek0210078h0002 (J.K.), 17ak0101068h0001 (J.K.), 17gm1010006s0101 (J.K.), and 18ck0106243h0003 (J.K.); the Ministry of Health, Labour and Welfare of Japan under grant number 15654110 (M.M.); the ONO Medical Research Foundation (J.K.). The funders had no role in study design, data collection and analysis, decision to publish, or preparation of the manuscript.

**Competing interests:** The authors have declared that no competing interests exist.

by 16S rRNA gene amplicon sequencing. However, processing these sequencing data requires a large amount of computational efforts. Post-sequencing the computational analysis [5] of microbial data broadly consists of three phases. (1) Data cleaning and normalization: comprised of multiple steps depending on the data source and sequencing technology, such as binning, pair-ends joining, and quality filtering; (2) Taxonomy and abundance estimation: in which taxonomy is assigned to the processed sequence reads, and their abundance in biological samples is estimated; and (3) analysis and interpretation of alpha and beta diversities, and functional annotation and the correlation between the microbial abundances and the physiological, environmental, or behavioral factors.

Both the first and the second phases above are sophisticated, time-consuming, and require high computational resources, in both 16S amplicon profiling and shotgun sequencing. QIIME [6] and Kraken [7] are well-known examples in this category. Other related tools include MEGAN [8], METAGEN-assist [9], EBI metagenomics, and MG-RAST [10].

In contrast, the third phase requires extensive user-interaction with researchers to select parameters and visualizing the output, especially when parameters with high dimensionality such as dietary, behavioral, and economic statuses are considered. The organization and storage of such multi-dimensional data types is challenging and a non-trivial task. MicrobiomeAnalyst [11], Calypso [12], Shiny-phyloseq [13], and Mian [14] are web-based online tools to address these challenges. Those tools provide interactive web interfaces to mediate R (such as phyloseq [15], vegan [16], and ade4 [17]) or Python packages. Although those well-designed tools provide various kinds of visualization and many sophisticated analytical approaches, they cannot store the data for sharing and reuse among project members. Moreover, some of those tools require the researchers to upload their data to third-party servers, which often invites data security concerns. When handling big multi-dimensional metadata, researchers often need to explore iteratively the efficacy of combining different parameters, or using different subsets of the data in the analytical framework. A database that can allow the users to manipulate stratified datasets quickly and efficiently would be extremely useful for such analysis.

We, therefore, aimed to develop a tool to facilitate the third phase of the analysis with the following features; (1) a smooth and interactive user interface to quickly and efficiently analyze the data with no programming efforts on the part of the user, (2) the ability to save the data in a readily accessible format, and (3) to be flexible and easily installed on individual workstations or servers to ensure quick access and secure data storage.

In this paper, we describe MANTA, a software program for creating an integrative database and analysis platform for microbiome and phenotypic data. MANTA has two important unique features: (1) the ability to store and share the data, either on-line or locally, in a user friendly easily-accessible database, and (2) providing an interactive environment to examine the correlation between the microbial abundances and other data collected such as dietary habits and lifestyle parameters, which can be of huge size and in multiple dimensions. MANTA is scalable, and further functionalities can be added as desired to the open-source code made available.

We have also demonstrated the usefulness of this platform by using a real-life dataset of microbiome and lifestyle-related data, which included dietary intake and physical activity obtained from 20 Japanese individuals. This case study shows that our platform can provide a novel hypothesis on the relationship between the relative abundance of specific bacteria and specific lifestyle parameters.

## Materials and methods

### Implementation

Our aim in this study was to develop a database framework that is able to store and share the data on human microbiome studies. The framework consists of a database and a web

application, including a suite of analytical and visualization tools; it provides analytical features via a graphical user interface that can easily facilitate visualizing and correlating microbiome and phenotypic data. MANTA-based instances can be accessed from any computer through a modern web browser.

We store all the microbiome and phenotypic data using PostgreSQL [18], an open-source relational database. The microbiome data need to be pre-processed and prepared in a standard format. In addition to microbiota composition, pre-processing was also performed to provide additional information. These additions included the identification of the dominating taxonomy for each sample, to allow for the plotting of easily readable bar charts or heat maps. This annotation was achieved by merging the low abundance taxonomies in the "others" category that was always set to be below a specific threshold (we used 10 percent in our application). Next, we added the alpha diversity indices, including Shannon, Simpson, and Chao1 [19–21]. Finally, we included the phylogenetic distances used for hierarchical clustering and principal coordinate analysis, such as Jaccard distance, weighted and unweighted UniFrac distance [22] or Bray-Curtis dissimilarity [23] for each pair of samples. These phylogenetic distances could be calculated using the R phyloseq, or vegan packages.

The phenotypic data can include—but are not limited to—multiple physical measurements and the measurements taken while exercising such as blood profiles, lifestyle questionnaires, and immunological studies. For convenience, we refer to these data as 'parameters'. The parameters were classified into continuous, nominal, and ordinal variables, and text. The data that are labeled as text type can only be browsed in the application and are not to be used for further analysis. To deal with variable sets of parameters, possibly from different studies, we designed database tables to store the parameters in the form of name-value pairs. The database schema is shown in the form of Entity-relationship (ER) diagram and is released together with the source code.

The user interface was developed using Google Web Toolkit, a Java-based framework for web application development [24]. All calculations were implemented in Java programming language.

## Case study: Correlations between dietary fat intake and microbiome

To demonstrate main functions of MANTA, we prepared an example database and named it MANTA demonstration database (MDD). The data stored in MDD is a subset of the NIBIOHN cohort data, a project conducted by National Institutes of Biomedical Innovation, health and Nutrition (manuscript in preparation). MDD includes twenty fecal samples collected from 20 healthy adult volunteers (21–41 years old, male) from Minamiuonuma City, Niigata Prefecture, Japan. The NIBIOHN cohort study also collected a wide range of parameters from the participants, including physical and exercise measurements, blood profiles, lifestyle questionnaires, and immunological parameters. (MDD includes only a subset of these parameters.) To enable quick access to these parameters, we further classified them into categories and subgroups (as listed in Table 1). Informed consent was obtained from all the participants. This study was approved by the Ethical Committee of National Institutes of Biomedical Innovation, Health, and Nutrition (KENEI-78).

The fecal samples were processed, and 16S rRNA gene amplicon sequencing was performed using Illumina MiSeq in the National Institutes of Biomedical Innovation, Health, and Nutrition, as described by Hosomi *et al.* [25]. The resulted sequences were analyzed using the QIIME software package [6]. The steps from trimming of paired-end reads to OTU picking were performed by QIIME Analysis Automating Script (Auto-q) [26, 27]. The pre-processed sequences were clustered into OTUs based on the sequence similarity ($>$ 97%) using open-reference OTU picking with UCLUST software [28] against the SILVA reference sequence library v128 [29, 30]. The taxonomy (phylum, class, order, family, and genus) and relative abundances

**Table 1. Main and subcategories of the parameters.**

| Class | Category | Subgroup |
|---|---|---|
| Information | Basic information | Basic information |
| Health condition | Medical history | Medical history |
| | | Family medical history |
| | | Presence of any malaise |
| | | Medication |
| | Menstruation | Menstruation |
| | Defecation habit | Defecation habit |
| Physical characteristics | Body composition | Body composition |
| | Blood profile | Blood profile |
| Lifestyle | Physical activity (accelerometer) | Physical activity |
| | Diet | Food Intake frequency |
| | | Amount of Food intake |
| | | Amount of Food class intake |
| | | Nutrients |
| | | Nutrition statistics |
| | | Eating behaviour |
| | Other lifestyle | Smoking |
| | | Physical activity (subjective) |
| | | Exercise habit |
| | | Working status |
| | | Sleep & rest |
| | | Stress and tiredness |

were calculated using the SILVA database [29] as the reference database. MANTA does not depend on any specific taxonomy systems but for MDD, we decided to use the NCBI Taxonomy Database [31] identifiers (taxon IDs). We converted the SILVA taxon names to the corresponding taxon IDs using TargetMine [32, 33]. We annotated the names that were not found in the NCBI Taxonomy Database as 'unclassified'. Although we have used QIIME in this case study, in principle, any suitable analysis tool can be used to provide the taxonomy and abundance data if prepared as per the guidelines on our website.

## Results

We have developed MANTA, a software program for creating an integrative database and analysis platform, that can store and correlate microbiome and phenotypic data. The platform is a web-based application and can be accessed with any modern web browser. The platform is designed for integrating and hosting a large quantity of data from multiple studies. MANTA can be installed on a local server. For users who wish to use the platform in PC or Mac, we also developed a stand-alone version (MANTA basic), which can be installed on a PC or Mac and provides a user interface to import the data with minimal effort (details described below). The source code and an example database (MANTA demonstration database, MDD) can be found at https://mizuguchilab.org/manta/. MDD demonstrates the main (but not all) functions of MANTA and does not provide user data upload and account management functions.

### Data import

The database schema of MANTA is available along with the source code. To build a new web-based database using MANTA, the user needs to pre-process the microbiome data and

phenotypic parameters to fit in the corresponding tables. In addition, the sample-to-sample distance, and the alpha-diversity should be calculated in advance.

MANTA basic offers smaller functionalities but provides a user interface for importing data more efficiently. The user can upload the data from the 'Data management' function using a graphical user interface. For more details, see S1 Appendix. The Jaccard distance and Bray-Curtis dissimilarity will be calculated instantly for the uploaded data.

## Data visualization and data analysis

The main page of the application shows a list of available samples in a scrollable table (Fig 1A). Above the table, there is a navigation bar that helps the user to navigate among the different views. On this sample list page, the user can browse all the details of an individual sample by clicking on the sample entry in the table. Since the primary purpose of this framework is to correlate the microbiota composition with the phenotypic data, the samples without microbiota data are shaded in grey and are not selectable for further analysis.

After a set of samples are chosen, an analysis screen with a few tabs appears (Fig 1B). Each tab represents a different start point for data visualization and data analysis. Currently, we propose three entry points as follows: 'Microbiota composition', 'Phenotypic parameters', and 'Compare two parameters'.

The first tab, 'Microbiota composition', shows a table of the microbiota data of a specific rank (Fig 1B). The default rank is 'phylum', and the user can change the rank using the drop-

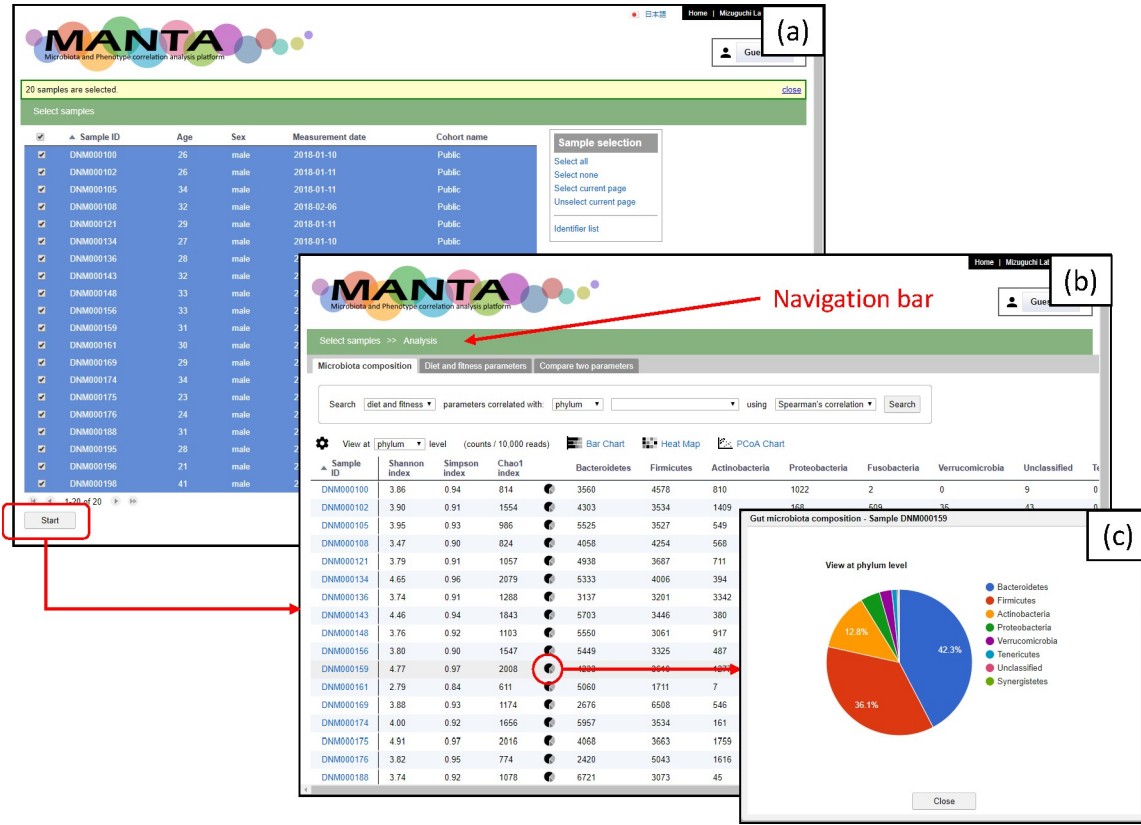

**Fig 1. General view of the application user interface.** (a) The main page of the application shows a list of available samples in a scrollable table. (b) The first tab of the analysis page, 'Microbiota composition', shows a table of the microbiota data of a specific rank. (c) Clicking on the Pie Chart icon displays the current microbiota composition for the current rank as a pie chart.

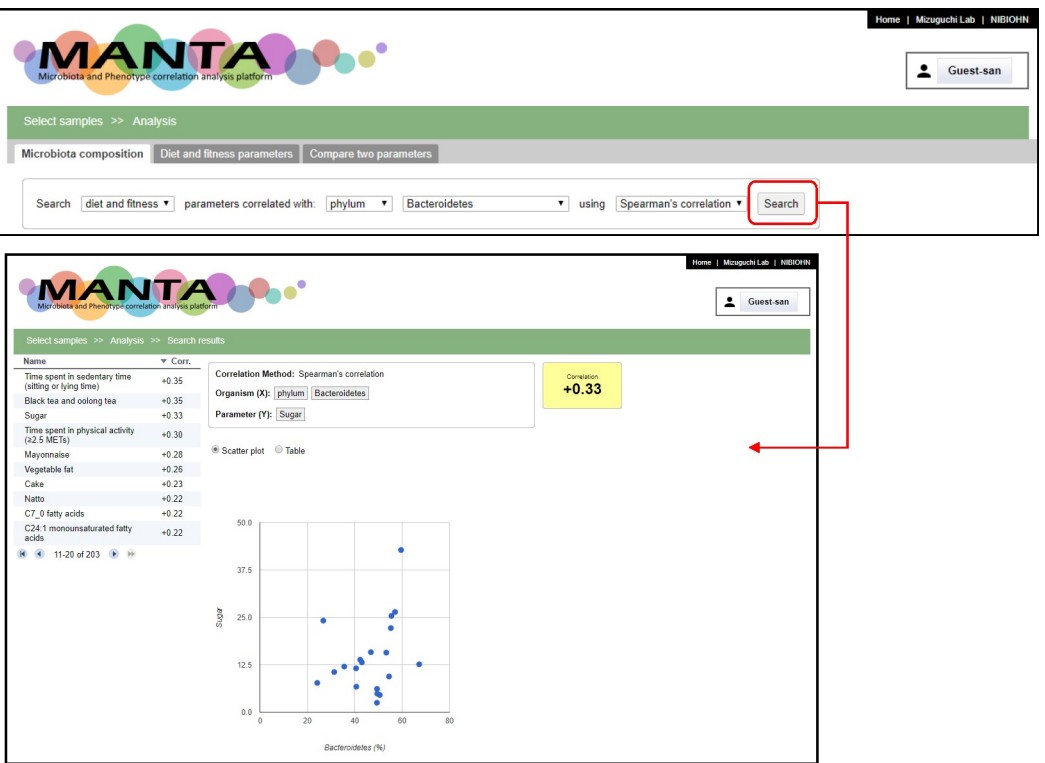

**Fig 2. Correlation search for *Bacteroidetes*.** The table in the left part of the search results shows the correlation coefficient in descending order. The user can toggle the order by clicking on the column header. The right part shows the scatter plot for *Bacteroidetes* relative abundance and the selected parameter (sugar intake in this example).

down list. If there are more than ten taxa, only the ten most abundant ones will be shown by default. The user can change the displayed taxa (columns) by clicking on the column management icon—the gear icon at the upper left. Each row contains three types of diversity indexes and a pie chart icon. Clicking on the pie chart icon displays the current microbiota composition for the current rank as a pie chart (Fig 1C). Clicking on the taxon expands the pie chart and shows the taxon composition of the next rank for the selected taxon.

At the beginning of the page, there is a box to compare the microbiota composition with the parameter measurements (limited to continuous variables). After choosing a taxon and clicking on the search button, the system will calculate the correlation between the selected microbial taxon and the available numeric parameters (Fig 2). Two types of correlation calculations are available, the Pearson's correlation coefficient and the Spearman's rank correlation coefficient, which is known to be more robust against outliers [34, 35]. The results are displayed as a table showing the obtained correlation coefficient in descending order. Clicking on the parameter name displays the correlation between 'Organism (x-axis)' and 'Parameter (y-axis)' in a scatter plot or table view. Three visualization options are available above the composition table: 'Bar Chart', 'Heat Map', and 'PCoA Chart' (Fig 3). The Bar Chart option displays the microbiota composition in a stacked composite bar chart plot, whereas the Heat Map option colors different taxa by proportion.

The hierarchical clustering feature allows the user to cluster the samples using three different linkage types, average, complete, and single. The clustering is based on the pre-calculated distances, for example, weighted and unweighted UniFrac [22] or Bray-Curtis dissimilarity [23], and the samples are sorted according to the dendrogram obtained by this clustering

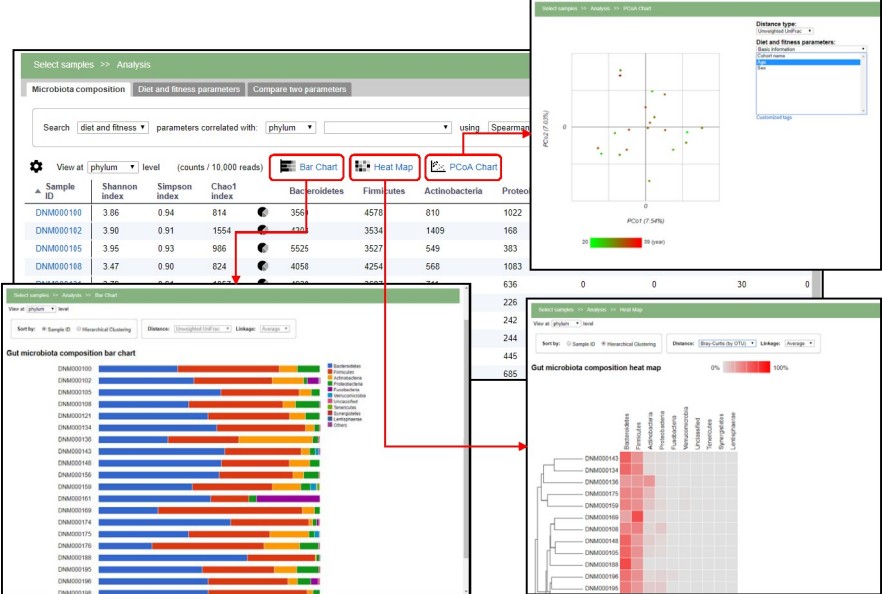

**Fig 3. Microbiome data visualization.** Clicking on the Bar Chart and Heat Map icons will display the microbiome data. The samples can be ordered by sample identifiers or hierarchical clustering. There are several options for the distance metric and linkages. The users can perform principal coordinate analysis (PCoA) by clicking on the PCoA Chart icon, display its result in a 2D scatter plot, and color the dot in the scatter plot according to the selected parameter from the drop-down list at the right side.

operation. The user can change the displayed rank using the drop-down menu at the top; the default rank is set to the one chosen on the previous (microbiota composition) page. Clicking on the taxon bar expands the bar chart visualization and shows the composition of the next rank of the selected taxon. The items displayed here are chosen according to the pre-calculated dominant taxa.

PCoA, also known as classical multi-dimensional scaling, is an analytical approach that visualizes distance matrix information in the form of a two-dimensional scatter plot. Four different distance metrics are available in the system, as described above. Each point in the PCoA plot represents a sample. The continuous or nominal parameters can be used to color the sample points (Fig 3), which can help to identify the correlation between microbiota composition and the chosen parameter.

The tab 'Phenotypic parameters' provides an alternative entry point for the data analysis (Fig 4A). The system calculates the correlation coefficient (Pearson's or Spearman's) for the selected parameter against the microbial taxonomies. The last tab provides a function to show the correlation coefficient of an arbitrary pair of designated taxa or the parameter (Fig 4B).

## Case study: Correlations between fat intake and microbiome

The microbiome data from 20 Japanese adults (21–41 years old, male) were analyzed as a case study to test the efficacy and usefulness of our tool. More information about the sample collection is described in the "Materials and methods" section. This example database, MDD, can be found at https://mizuguchilab.org/manta/mdd/. First, we performed PCoA using Bray-Curtis based on OTUs as distance type to evaluate the similarity among the volunteers. The 1$^{st}$ and 2$^{nd}$ principal coordinates explained 21.24% and 12.43% of the variance, respectively (Fig 5). For this plot, the user can change the color of the dots according to the selected 'Diet and physical activity parameters' by clicking on one of the parameters of interest from the pull-down

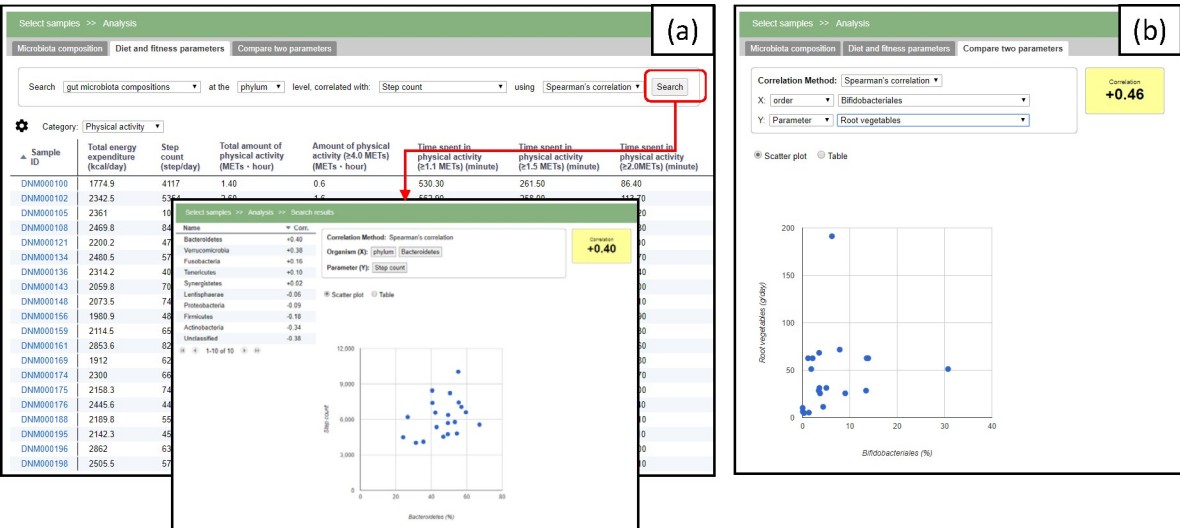

**Fig 4. Screenshots of the other two tabs.** (a) The phenotypic parameters can be browsed in the 'Phenotypic parameters' tab. The user can examine the correlations to the microbiota composition for a selected parameter. (b) In the 'Compare two parameters' tab, the correlation of any combination of the taxa or parameter is calculated and also displayed in a scatter plot.

menus. This function led us to find that the volunteers were grouped separately according to several specific parameters such as 'Cooking oil', 'Fat', or 'ω-6 polyunsaturated fatty acids' (Fig 5A, 5B and 5C, respectively). Next, we searched for the gut microbiota compositions that correlated with 'Fat' based on the Spearman's correlation coefficient. This analysis showed that *Lachnospiraceae* had a positive correlation with fat intake, estimated using a brief-type self-administered diet history questionnaire (Fig 6).

*Lachnospiraceae* comprises butyrate producers, and it was reported that a high-fat diet with low carbohydrate intake is associated with the abundance ratios of *Firmicutes* to *Lachnospiraceae* and *Ruminococcaceae* [36]. With these results, we hypothesized that the ratios of *Lachnospiraceae* to *Ruminococcaceae* were affected mainly by diet, and especially fat intake.

We then explored the dietary and physical activity parameters that correlated with *Lachnospiraceae*. We observed a positive correlation between this group and monounsaturated fatty acid or saturated fatty acid intake (Fig 7A), and a negative correlation with parameters related to the time spent doing physical activity (Fig 7B). Interestingly, *Ruminococcaceae* showed a similar but notably distinct tendency. This family showed a positive correlation with ω-6 polyunsaturated fatty acid intake (Fig 7C) and a negative correlation with parameters related to body composition, such as body weight and total energy expenditure, as well as the intake of carbohydrate sources such as boiled rice and grains (Fig 7D). Since *Lachnospiraceae* showed no strong correlation with the parameters that correlated with the *Ruminococcaceae* relative abundance, and vice versa, it was suggested that these two microbes are independently affected by diet.

As shown in the case study, our tool allowed us to hypothesize the relationship between gut microbiota composition (*Lachnospiraceae* or *Ruminococcaceae*) and diet and physical activity parameters (fatty acids, physical activity, or body composition). Our results are consistent with those of Zhang *et al.* [36], who suggested that different types of fatty acids independently affect *Lachnospiraceae* and *Ruminococcaceae*. Although these findings will require further verification for a deeper understanding of the relevant relationships, it is noteworthy that our tool successfully hypothesized probable links between microbiome and lifestyle.

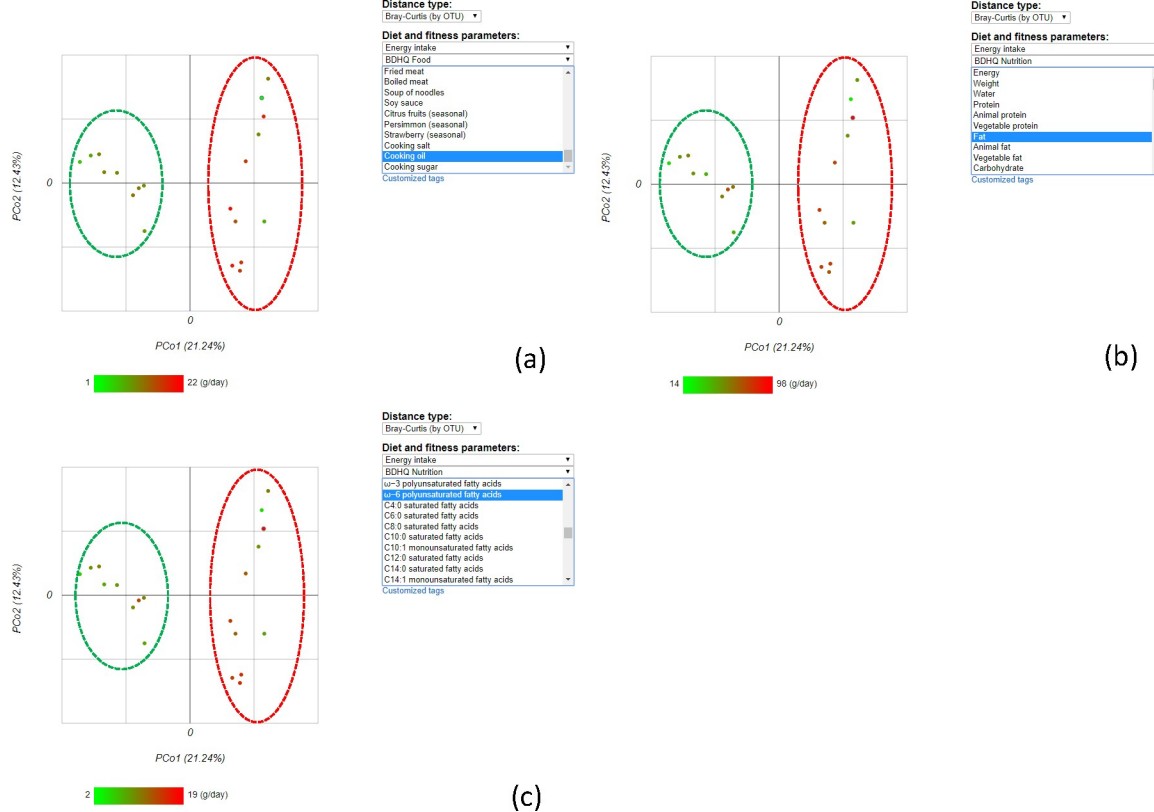

**Fig 5. Relationship between microbiome composition and fat intake as detected by Principal Coordinate Analysis (PCoA).** The dots are colored by the energy intake from (a) cooking oil, (b) fat, and (c) ω-6 polyunsaturated fatty acids. The coloring suggests that the participants could be separated into two groups, namely, High-fat (red dashed circle) and low-fat consumption (green dashed circle).

## Discussion

To fully understand how the microbiota affect lifestyle and vice versa, it is essential to collect microbiome data together with a detailed information about the host or the environment, in which the microbiome data were obtained. However, the analysis of such heterogeneous data is a non-trivial task. So far, the web applications that may easily allow the users to browse and analyze such data are not publicly available. Therefore, we developed a software platform for microbiome studies that allows the creation of an integrative database of the microbiome and phenotypic data and provides a user-friendly interface with online analytical functions to uncover the relationships between bacterial composition and phenotypic features.

Our case study suggested that MANTA is not only adept at storing microbiome data but is also capable of clearly demonstrating the correlations between the microbiome and lifestyle parameters. MANTA is user friendly and much of the operations can be performed by the user with only a few clicks of the mouse. Although we have analyzed the data from the human gut microbiome as the case study, MANTA framework can easily be used to investigate microbiome data from non-human hosts.

While MANTA can accommodate different data structures, it requires significant efforts by the database administrator to format the data into specific tables. To address this issue, we have developed MANTA basic, which contains a smaller number of database tables (and hence, slightly limited capabilities) but has retained the core MANTA functionality. In addition, MANTA basic is bundled with a portable web server and hence requires no additional

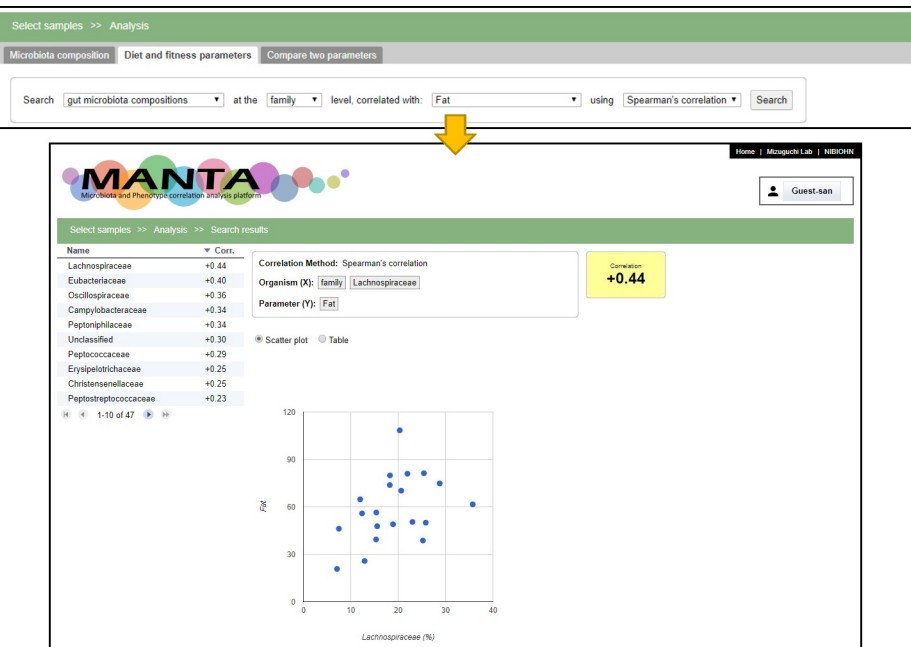

**Fig 6. Search for microbiota that correlates with the parameter 'Fat'.** Our tool enables investigation of the correlation between 'microbiota composition' and 'Diet and physical activity parameters' by a simple operation (selecting parameters of interest from the drop-down list). The tool shows that *Lachnospiraceae* was the family most positively correlated with fat intake.

applications to be installed separately. A comparison of MANTA and MANTA basic is listed in Table 2. In MANTA basic, we have implemented a data management interface to allow the user to upload their data; the user only needs to prepare two file types, microbiota and phenotypic parameter data in the form of tab-delimited text. Further details on how to use MANTA basic can be found in the online documentation (https://mizuguchilab.org/manta/manta-basic.html). Both MANTA and MANTA basic are available at https://mizuguchilab.org/manta/.

We are in the process of expanding our global collaborations to elucidate associations between microbiota and the host using a variety of cohorts; thus, data will be provided by various users, and a more comprehensive user administration function will be necessary. In addition, a well-organized ontology is required to integrate the phenotypic descriptions from different cohorts. Moreover, some studies may collect data from multiple time points and thus necessitating an ability to perform temporal analysis as required. We will continuously develop new features to address these and other emerging issues in microbiome research.

## Conclusions

MANTA is an analysis platform that can assist researchers working on human microbiome studies with data sharing and analysis, either on-line or on their desktop. The focus on data storage and sharing implies that MANTA is not designed to replace other tools, for example, for processing the raw sequencing data. However, MANTA is more than a generic database tool and provides some specialist functionalities; currently our emphasis is to examine the correlation between the microbial abundances and parameters such as dietary or life style parameters. MANTA addresses a long-standing challenge in

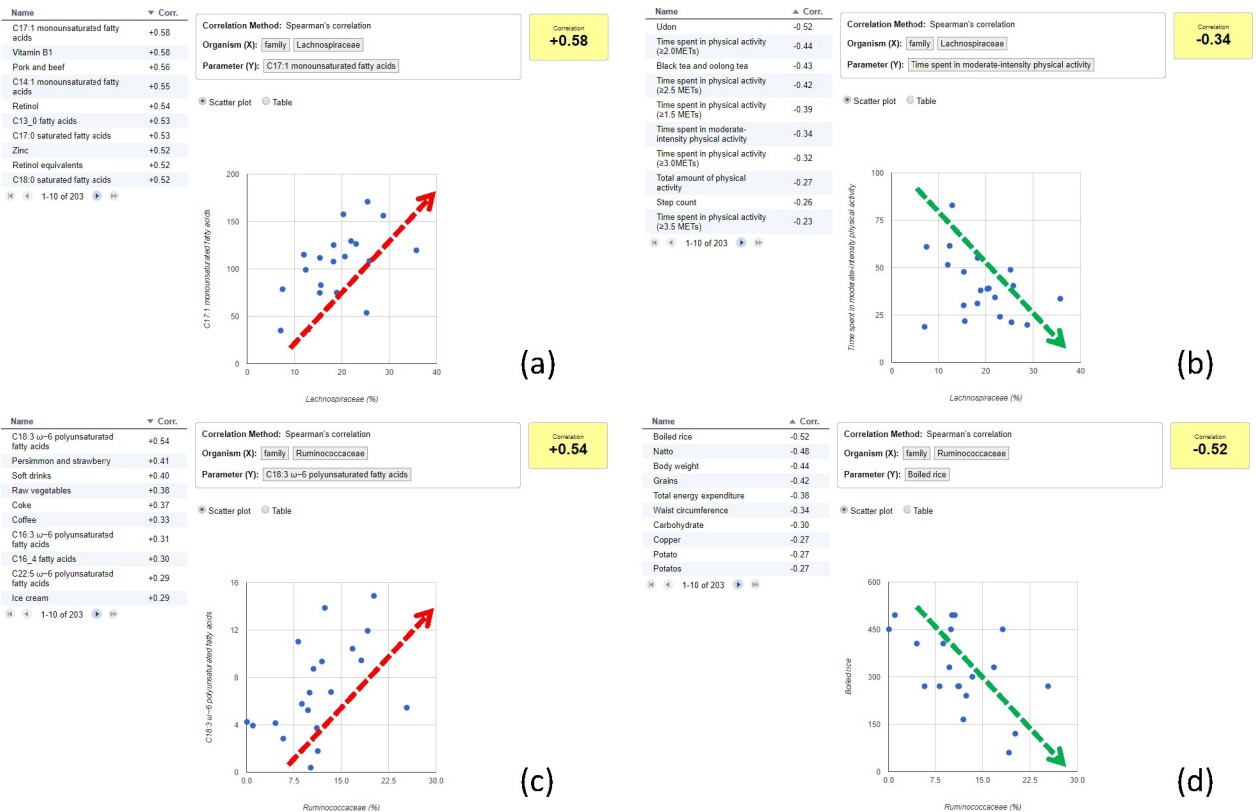

**Fig 7. Search of 'diet and physical activity parameters' that correlate with *Lachnospiraceae* or *Ruminococcaceae*.** (a) Positive correlation between monounsaturated fatty acids (MUFAs) and *Lachnospiraceae*. The table (left) shows the top 10 positively correlated parameters. The scatter plot (right) shows the relationship between *Lachnospiraceae* (x-axis) and C17:1 monounsaturated fatty acids (y-axis), which exhibited a positive correlation. (b) Negative correlation between physical activity and *Lachnospiraceae*. The table (left) shows the top 10 negatively correlated parameters. The scatter plot (right) shows the relationship between *Lachnospiraceae* (x-axis) and time spent in moderate-intensity physical activity (y-axis), which showed a negative correlation. (c) Positive correlation between polyunsaturated fatty acids (PUFA) and *Ruminococcaceae*. The table (left) shows the top 10 positively correlated parameters. The scatter plot (right) shows the relationship between *Ruminococcaceae* (x-axis) and C18:3 ω-6 polyunsaturated fatty acids (y-axis), which showed a positive correlation. (d) Negative correlation between carbohydrates (e.g. boiled rice) and *Ruminococcaceae*. The table (left) shows the top 10 negatively correlated parameters. The scatter plot (right) shows the relationship between *Ruminococcaceae* (x-axis) and 'Boiled rice' (y-axis), which exhibited a negative correlation.

microbiome research and not so easily achievable by other tools that are currently available. The MANTA framework has the potential to adequately assist studies involving human data as well those from other organisms.

**Table 2. A comparison of MANTA and MANTA basic.**

| Features | MANTA | MANTA basic |
|---|---|---|
| Visualization (Heat map, Bar chart, Pie chart) | Yes | Yes |
| Hierarchical clustering | Yes | Yes |
| Correlation analysis | Yes | Yes |
| PCoA | Yes | Yes |
| Parameter grouping* | Yes | No |
| User login | Yes | No |
| Data upload interface | No | Yes |

* The grouping here means there is no categories or subgroup for the parameters like we described in Table 1.

## Supporting information

**S1 Appendix. An illustration for importing data into MANTA basic.**
(DOCX)

## Acknowledgments

We thank the members in the Mizuguchi lab for the critical reading of the manuscript.

## Author Contributions

**Conceptualization:** Yi-An Chen, Jonguk Park, Kenji Mizuguchi.

**Data curation:** Yi-An Chen.

**Formal analysis:** Yayoi Natsume-Kitatani, Attayeb Mohsen.

**Funding acquisition:** Koji Hosomi, Haruka Murakami, Motohiko Miyachi, Jun Kunisawa, Kenji Mizuguchi.

**Investigation:** Hitoshi Kawashima, Koji Hosomi, Kumpei Tanisawa, Harumi Ohno, Kana Konishi.

**Methodology:** Yi-An Chen, Jonguk Park.

**Project administration:** Kenji Mizuguchi.

**Software:** Yi-An Chen.

**Supervision:** Haruka Murakami, Motohiko Miyachi, Jun Kunisawa, Kenji Mizuguchi.

**Visualization:** Yi-An Chen.

**Writing – original draft:** Yi-An Chen, Yayoi Natsume-Kitatani.

**Writing – review & editing:** Kenji Mizuguchi.

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
