## [Decision Letter · Decision Letter 0]

13 Jul 2020

PONE-D-20-09945

MANTA, an integrative database and analysis platform that relates microbiome and phenotypic data

PLOS ONE

Dear Dr. Chen,

Thank you for submitting your manuscript to PLOS ONE. After careful consideration, we feel that it has merit but does not fully meet PLOS ONE’s publication criteria as it currently stands. Therefore, we invite you to submit a revised version of the manuscript that addresses the points raised during the review process.

We look forward to receiving your revised manuscript.

Kind regards,

Lingling An

Academic Editor

PLOS ONE

Journal Requirements:

Reviewers' comments:

Reviewer's Responses to Questions

**Comments to the Author**

1. Is the manuscript technically sound, and do the data support the conclusions?

Reviewer #1: Partly

Reviewer #2: Yes

Reviewer #3: Partly

2. Has the statistical analysis been performed appropriately and rigorously? 

Reviewer #1: N/A

Reviewer #2: N/A

Reviewer #3: N/A

3. Have the authors made all data underlying the findings in their manuscript fully available?

Reviewer #1: Yes

Reviewer #2: Yes

Reviewer #3: No

4. Is the manuscript presented in an intelligible fashion and written in standard English?

Reviewer #1: No

Reviewer #2: Yes

Reviewer #3: Yes

5. Review Comments to the Author

Reviewer #1: This reviewer is not a bioinformatician by training, but has basic understanding. Therefore, the comments are more from a end-user perspective.

The article requires substantial rewriting. The reviewer had to jump across manuscript to find and link information.

Should follow the pattern of guiding the readers from what the tool is about to how one can use it properly.

The introduction is vague and requires restructuring. It can be importance of microbiota, challenges in microbiota data, currently available tools and what MANTA does in more succinct way.

MANTA provides an alternative to currently available tools. However, lacks functionality commonly found in tools that were cited in the paper. There are many tools (MicrobiomeAnalyst or Calypso, shiny-phyloseq) which can do more than MANTA. A comparison table of MANTA functionality and other tools the authors mention will be useful.

The concept to database is a bit far fetched in this study. As far as I see, it is like other tools e.g. MicrobiomeAnalyst or Calypso where you just upload tables and do analysis. It is somewhat similar to a phyloseq object only that MANTA is implemented in java and stores info in *.db which can have its advantages. However, there is a need to incorporate phylogenetic and sequence information.

I would also like the authors to check the analysis options that are provided by tool called Main (still testing phase). https://www.biorxiv.org/content/10.1101/416073v1

For MANTA to have application for broader analysis, it should have provision for making database using ASV sequences. In this way you can compare across studies to identify common ASVs for analysis. OTU-taxonomy based approach is not recommended (Callahan et al. 2017 PMID: 28731476). For e.g. if I have first project for cancer and healthy microbiota for which I have ASVs sequence abundances and another project for IBD and healthy microbiota, I should be able to analyse these together for a meta analysis. This would be the advantage of having a database like structure. The advantage will be that the web server of authors may not be overloaded by analyzing raw reads but still has high utility because sequence information is stored for every project. This can be organized at user level where only specific user who owns the data can access all their projects that have been processed in similar way.

For research groups that do not have data scientists, they can still do basic analysis that MANTA provides. What they need is advanced analysis with well written documentation and tutorials to guide such analysis. e.g Machine learning random forests etc.

Minor comments:

Through out the article please use the term microbiome and microbiota appropriately, see https://microbiomejournal.biomedcentral.com/articles/10.1186/s40168-015-0094-5.

In addiiton, the name of the tool can be changed to its betterment as the name MANTA has been used previously.

manta http://www.bioconductor.org/packages/release/bioc/html/manta.html

manta: https://msystems.asm.org/content/5/1/e00903-19

In summary, I see great potential for the tool but it requires substantial improvements for having a wider impact or usefulness.

Reviewer #2: This manuscript provides a well-designed website to analyze microbiome datasets. The key idea is to provide a graphical interface to enable users to browse or analyze their own data or the example datasets. To my understanding, this work has overlapping features with MicrobiomeAnalyst. Thus the validity of the innovation is not clear. I have also listed other issues as follows.

1. The website (https://mizuguchilab.org/manta-example/#en_heatmap) does not function well (Mac Mojave). It always shows “An error occurred while attempting to contact the server. Please check your network connection and try again.” I cannot test this database as a reviewer.

2. The online account system is probably incomplete. Users cannot register. This limits the use of MANTA website to only its developers.

3. I am not clear which function is uniquely available for MANTA and not in other software/websites.

4. The authors claims that MANTA is ‘an integrative database’. As database should allow users to create, read, update, or delete data entries, I do not see MANTA currently support these functions in its website.

5. For ordinal parameters, how the correlation is calculated and visualized?

6. “The assigned ranks and names were searched against the TargetMine data warehouse… ” TargetMine seems to be a gene analysis website developed by the same first author. I am not clear how this website can be used for searching microbiome names.

7. There is no download link for “MANTA basic”. I cannot find the download link and cannot test it either.

Reviewer #3: The manuscript describes a website, MANTA, that can be used to store user-uploaded information and perform analysis through web-based user interface. That being said, I encountered some difficulties when I was trying this website and cannot evaluate whether the website is good-to-go or not (will be described below). So I mostly comment on the manuscript itself, trying to guess what the authors were trying to do and how it should look like. Below please find my comments.

1. This is about the website itself. When I want to try this website by clicking on the link in the example dataset tab (https://mizuguchilab.org/manta-example/), the system popped up a message saying that “An error occurred while attempting to contact the server. Please check your network connection and try again”. Due to this I cannot evaluate the full functionality of this website. Please see attached Figure 1 for a screenshot.

2. From the description of the manuscript it looks like the users can upload some information and conduct some analysis by themselves. The tutorial also offers such helps. I however do not see the same figure as shown in the tutorial, in which users are supposed to upload their own data through “Data management” menu items but I just cannot see such thing exists (see attached Figure 2). Two possible reasons may be that the website is down when I was trying to test it, or that there are certain problems associated with the website. I suggest the authors carefully check their websites before submitting their manuscript.

3. The manuscript claims that it is able to relate microbiome and phenotypic data. Yet the functionality that they supported is simply invoking existing R commands/libraries on the datasets without providing novel insights. Furthermore, while the authors claim that they want to develop a website for people not familiar with running programs on command line-based platforms, they still require the microbiota to be analyzed and the OTUs extracted before submitting the data to the MANTA website. This creates a dilemma—researchers that can run the microbiota analysis by themselves do not need their website for phenotypic correlation analysis, and that people who do not know how to run commands cannot analyze microbiota as the authors require. I hence doubt the very existence meaning of the MANTA website and hope that the authors can very carefully think and discuss with non-computational scientists about their needs.

4. The authors are mixing up 16S-based methods with whole-metagenome-based methods, as seen when they are talking about tools such as QIIME, MEGAN, Kraken, and Metaphlan2. The web-based systems that they talked about are mostly whole-metagenome-based methods as well. I suggest the authors at least distinguishing to their best effort between 16S and whole-metagenome and/or discussing their potential uses toward their own systems.

5. There is not “Data preprocessing section” as mentioned in line 96.

6. I am not very sure why the authors mention “free text” in line 101 as one of their parameter forms. I frankly do not think their system can deal with free text according to their description of the website functionalities. Maybe they just want to raise a possible example. If this is the case then this part needs to be re-written in order not to confuse the readers.

7. The authors talked about converting absolute abundances to relative ones for analysis purpose (line 108). Pardon me, but to the best of my knowledge common software packages provide mostly proportions. Therefore I have no idea why the authors want to claim this function.

8. For the description of the case study in Methods, the authors should provide more details, including when the 20 Japanese adults study was conducted. The authors may also want to explain why only a subset of parameters was released. (hence that perhaps not all data are publicly available for publication or review purpose)

9. The analysis pipeline using QIIME was not up-to-date. I understand that some people still use QIIME instead of QIIME2 since they are quite different in some sense; however the SILVA database is outdated. I also don’t understand what the authors mean by mentioning “the taxonomy hierarchy is continuous refined and updated…we reconstructed the phylogenetic lineages using the NCBI taxonomy database (line 143-145).” This part is vague at best, and I honestly do not know what their purposes to “reconstruct” the SILVA-based taxonomic inferences are. Please check your pipeline and conduct the analysis using the most up-to-date datasets/software/versions.

10. The authors mention that the parameters that they are able to asses are “limited to continuous variables (line 200)”. I wonder why they set such restrictions. Does it also mean that their system cannot handle nominal or categorical data?

11. I highly suggest the authors check QIIME2 visualization support since a lot of functionalities that they claim are already supported by QIIME2, for example the coloring of sample points according to the numerical or nominal parameters (line 238).

12. The resolutions of the figures are very poor. Please consider making higher quality ones.

13. Due to poor figure quality and the inability to test the website, I cannot tell whether the authors provide significance metrics such as p-value/FDR or adjust R-squared. Please consider adding such metrics in order to help your users interpreting data if these are not supported.

14. The authors claim that “for users who wish to use the platform for analyzing their data, we also developed MANTA basic.” This is also very intriguing since it hinted that the existing MANTA website only supports the display of the 20-Japanese data, and that the users need to install their own system if they want to analyze their data. If this is true then I honestly do not know the existence meaning of the MANTA website. Please clarify if this is the case.

15. I don’t understand what does “parameter grouping” mean during the comparison between MANTA and MANTA Basic. Please clarify.

16. (line 173) pageable  do you mean scrollable?

17. (line 192) 10 most abundant will  10 most abundant ones will

6. PLOS authors have the option to publish the peer review history of their article (what does this mean?). If published, this will include your full peer review and any attached files.

Reviewer #1: No

Reviewer #2: No

Reviewer #3: No

---

## [Author Response · Author response to Decision Letter 0]

26 Aug 2020

Response to Reviewers

To present the aim of our work clearly, we have reorganized the introduction. Besides, we also refined several paragraphs that may have led to some misunderstandings. Concerning the failure of the website, we have fixed the problem and started to monitor the availability of the service.

Details: 

Reviewer #1: This reviewer is not a bioinformatician by training, but has basic understanding. Therefore, the comments are more from a end-user perspective.

The article requires substantial rewriting. The reviewer had to jump across manuscript to find and link information.

Should follow the pattern of guiding the readers from what the tool is about to how one can use it properly.

The introduction is vague and requires restructuring. It can be importance of microbiota, challenges in microbiota data, currently available tools and what MANTA does in more succinct way.

Thank you for the comments. We have reorganized the introduction following the reviewer’s suggestions. We rewrote it to reflect precisely the purpose and the aim of our work, and we believe that the current changes have increased the clarity of our article.

MANTA provides an alternative to currently available tools. However, lacks functionality commonly found in tools that were cited in the paper. There are many tools (MicrobiomeAnalyst or Calypso, shiny-phyloseq) which can do more than MANTA. A comparison table of MANTA functionality and other tools the authors mention will be useful.

The concept to database is a bit far fetched in this study. As far as I see, it is like other tools e.g. MicrobiomeAnalyst or Calypso where you just upload tables and do analysis. It is somewhat similar to a phyloseq object only that MANTA is implemented in java and stores info in *.db which can have its advantages. However, there is a need to incorporate phylogenetic and sequence information.

I would also like the authors to check the analysis options that are provided by tool called Main (still testing phase). https://www.biorxiv.org/content/10.1101/416073v1

MANTA is a software program that can assist researchers working on human microbiome studies with data sharing and analysis, either on-line or on their desktop. The focus of the data storage and sharing means that MANTA is not designed to replace other tools, for example, for processing the raw sequencing data. However, it is beyond a generic database tool and provides some specialist functionalities; currently our emphasis is on the ability to examine the correlation between the microbial abundances and other parameters such as dietary or life style parameters, an arbitrary (large) number of them that are being collected by the study. We chose to implement this function in MANTA because it is typically requested by our collaborators and yet it is not so easily achievable by other tools that are currently available. For some other analysis functions (such as phylogenetic distances), we may have to rely on external tools, and we have now described those tools including R phyloseq, or vegan package in the implementation section of the materials and methods. Still, MANTA’s flexible structure allows us to add new functionalities easily, and we plan to do so in the future versions.

For MANTA to have application for broader analysis, it should have provision for making database using ASV sequences. In this way you can compare across studies to identify common ASVs for analysis. OTU-taxonomy based approach is not recommended (Callahan et al. 2017 PMID: 28731476). For e.g. if I have first project for cancer and healthy microbiota for which I have ASVs sequence abundances and another project for IBD and healthy microbiota, I should be able to analyse these together for a meta analysis. This would be the advantage of having a database like structure. The advantage will be that the web server of authors may not be overloaded by analyzing raw reads but still has high utility because sequence information is stored for every project. This can be organized at user level where only specific user who owns the data can access all their projects that have been processed in similar way.

MANTA can be used for either ASVs or OTUs, with no difference. In our case study, we showed an example, which happened to have used OTUs. We have revised the paragraph to clarify this concept; we have also added further clarification in the section related to MANTA’s functions to avoid this confusion.

For research groups that do not have data scientists, they can still do basic analysis that MANTA provides. What they need is advanced analysis with well written documentation and tutorials to guide such analysis. e.g Machine learning random forests etc.

We are aware that users competent with command-line tools could do the same analysis using other tools. However, our typical use case scenarios will involve an interactive examination of a few thousand parameter combinations, especially dealing with diverse data types such as dietary habits, which can consist of hundreds of measurements. Sharing both the original data and the analysis results among all the project members would be a complicated task and instead, we believe that a tool such as MANTA can do a better job by allowing all the members to browse and examine the data.

Minor comments:

Through out the article please use the term microbiome and microbiota appropriately, see https://microbiomejournal.biomedcentral.com/articles/10.1186/s40168-015-0094-5.

Thank you for raising this issue. We have replaced all the occurrences of the term to be compatible with the suggestions in this article.

In addition, the name of the tool can be changed to its betterment as the name MANTA has been used previously.

manta http://www.bioconductor.org/packages/release/bioc/html/manta.html

manta: https://msystems.asm.org/content/5/1/e00903-19

We appreciate the information but we wish to retain the name MANTA, because we released the project on GitHub in March 2019 (earlier than the resources above), and we have already presented the work in several conferences including ISMB 2019 (Intelligent Systems for Molecular Biology). In addition, several microbiome projects have adopted MANTA to host their data.

In summary, I see great potential for the tool but it requires substantial improvements for having a wider impact or usefulness.

We thank the reviewer for their constructive and critical feedback.

Reviewer #2: This manuscript provides a well-designed website to analyze microbiome datasets. The key idea is to provide a graphical interface to enable users to browse or analyze their own data or the example datasets. To my understanding, this work has overlapping features with MicrobiomeAnalyst. Thus the validity of the innovation is not clear. I have also listed other issues as follows.

1. The website (https://mizuguchilab.org/manta-example/#en_heatmap) does not function well (Mac Mojave). It always shows “An error occurred while attempting to contact the server. Please check your network connection and try again.” I cannot test this database as a reviewer.

We apologize for the service having been temporarily unavailable. We have fixed the problem and it is fully functional now. We have also started monitoring the service to prevent any connection failure.

2. The online account system is probably incomplete. Users cannot register. This limits the use of MANTA website to only its developers.

The example database MANTA demonstration database (MDD) is a demonstration of main (but not all) functions of MANTA; it does not provide user data upload and account management functions. We clarified this point by adding notes on both the manuscript and the website.

3. I am not clear which function is uniquely available for MANTA and not in other software/websites.

MANTA has two important unique features: 1) the ability to store and share the data, either on-line or locally, in a user friendly easily-accessible database, and 2) providing an interactive environment to examine the correlation between the microbial abundances and other data collected such as dietary habits and lifestyle parameters, which can be of huge size and in multiple dimensions. We have made these two unique features mentioned clearly in the manuscript.

4. The authors claims that MANTA is ‘an integrative database’. As database should allow users to create, read, update, or delete data entries, I do not see MANTA currently support these functions in its website.

At the moment, the data management in MANTA relies on the database management system in PostgreSQL database. To transform MANTA into a more mature data storage/sharing system, we will work on adding functions for data manipulation within the system.

5. For ordinal parameters, how the correlation is calculated and visualized?

At the moment, only the continuous variables are used for the correlation analysis. We plan to make use of the ordinal parameters for analysis in the future development.

6. “The assigned ranks and names were searched against the TargetMine data warehouse… ” TargetMine seems to be a gene analysis website developed by the same first author. I am not clear how this website can be used for searching microbiome names.

TargetMine is an integrated data warehouse for target prioritization. It integrates, among many other data types, taxonomy information. In building our example database, we used TargetMine to determine the taxonomy identifiers for our data but users can use any tool for preparing their data. We have clarified this point in the manuscript.

7. There is no download link for “MANTA basic”. I cannot find the download link and cannot test it either.

Thank you for telling us the difficulty in finding the download link. Maybe we did not design our website very well. We have reorganized it and we hope these changes can increase the visibility of the download link.

Reviewer #3: The manuscript describes a website, MANTA, that can be used to store user-uploaded information and perform analysis through web-based user interface. That being said, I encountered some difficulties when I was trying this website and cannot evaluate whether the website is good-to-go or not (will be described below). So I mostly comment on the manuscript itself, trying to guess what the authors were trying to do and how it should look like. Below please find my comments.

1. This is about the website itself. When I want to try this website by clicking on the link in the example dataset tab (https://mizuguchilab.org/manta-example/), the system popped up a message saying that “An error occurred while attempting to contact the server. Please check your network connection and try again”. Due to this I cannot evaluate the full functionality of this website. Please see attached Figure 1 for a screenshot.

We apologize for the service having been temporarily unavailable. We have fixed the problem and it is fully functional now.

2. From the description of the manuscript it looks like the users can upload some information and conduct some analysis by themselves. The tutorial also offers such helps. I however do not see the same figure as shown in the tutorial, in which users are supposed to upload their own data through “Data management” menu items but I just cannot see such thing exists (see attached Figure 2). Two possible reasons may be that the website is down when I was trying to test it, or that there are certain problems associated with the website. I suggest the authors carefully check their websites before submitting their manuscript.

We have presented an analysis platform named MANTA, which can assist building an online data storage and sharing system for microbiota composition and various types of phenotypic data. Recognizing the amount of tasks for pre-processing the data and setting up an on-line database system, we have also developed a simplified tool named MANTA basic. MANTA basic can be installed and run on a PC/Mac and also provides a data importing interface (the “Data management” function).

3. The manuscript claims that it is able to relate microbiome and phenotypic data. Yet the functionality that they supported is simply invoking existing R commands/libraries on the datasets without providing novel insights. Furthermore, while the authors claim that they want to develop a website for people not familiar with running programs on command line-based platforms, they still require the microbiota to be analyzed and the OTUs extracted before submitting the data to the MANTA website. This creates a dilemma—researchers that can run the microbiota analysis by themselves do not need their website for phenotypic correlation analysis, and that people who do not know how to run commands cannot analyze microbiota as the authors require. I hence doubt the very existence meaning of the MANTA website and hope that the authors can very carefully think and discuss with non-computational scientists about their needs.

We have rewritten the Introduction and tried to define what we would like to achieve in the manuscript. 

Yes, visualization and correlation analysis could be done using R. However, unlike processing the sequencing data, which is usually a one-time task, this type of analysis requires the examination of a large number of parameters in a repeated manner, a tedious job to do using the command line tools. MANTA allows the users to reproduce and share their analysis results easily.

4. The authors are mixing up 16S-based methods with whole-metagenome-based methods, as seen when they are talking about tools such as QIIME, MEGAN, Kraken, and Metaphlan2. The web-based systems that they talked about are mostly whole-metagenome-based methods as well. I suggest the authors at least distinguishing to their best effort between 16S and whole-metagenome and/or discussing their potential uses toward their own systems.

Thank you for pointing this out. We have refined the introduction and explained that different tools are used for different purposes. 

5. There is not “Data preprocessing section” as mentioned in line 96.

We apologize for this mistake. After reorganizing the article, there is no “Data preprocessing section” any more. We have removed the phrase.

6. I am not very sure why the authors mention “free text” in line 101 as one of their parameter forms. I frankly do not think their system can deal with free text according to their description of the website functionalities. Maybe they just want to raise a possible example. If this is the case then this part needs to be re-written in order not to confuse the readers.

We apologize for the confusion. The “free text” here means that the value is a character string, and will not be used for any calculation. We have removed the word “free” in the manuscript.

7. The authors talked about converting absolute abundances to relative ones for analysis purpose (line 108). Pardon me, but to the best of my knowledge common software packages provide mostly proportions. Therefore I have no idea why the authors want to claim this function.

We agree that the use of the term "absolute" is not appropriate here. What we meant here was the counts before rarefication. We have updated the related paragraph by removing the confusing sentence.

8. For the description of the case study in Methods, the authors should provide more details, including when the 20 Japanese adults study was conducted. The authors may also want to explain why only a subset of parameters was released. (hence that perhaps not all data are publicly available for publication or review purpose)

This example database (now called MDD, to make its purpose clear) was developed to demonstrate main functions of MANTA. We took a small subset, sufficient for this purpose, of the data collected in a cohort study involving a few thousand healthy Japanese adults. Details of the original study will be described elsewhere and the whole data will be released with a new publication. In this manuscript, we have provided necessary information to understand this example database. The whole case study was conducted using only this subset of the data. We have clarified this point in the manuscript.

9. The analysis pipeline using QIIME was not up-to-date. I understand that some people still use QIIME instead of QIIME2 since they are quite different in some sense; however the SILVA database is outdated. I also don’t understand what the authors mean by mentioning “the taxonomy hierarchy is continuous refined and updated…we reconstructed the phylogenetic lineages using the NCBI taxonomy database (line 143-145).” This part is vague at best, and I honestly do not know what their purposes to “reconstruct” the SILVA-based taxonomic inferences are. Please check your pipeline and conduct the analysis using the most up-to-date datasets/software/versions.

We understand that the QIIME and SILVA versions used in the manuscript are somewhat old, but our objective here was to demonstrate the main functionality of MANTA, which is unaffected by the version numbers.

Regarding the “reconstruct” description, what we would like to explain was that the genus identified from the SILVA database was converted to taxonomy ID and we used the taxonomy hierarchy in the NCBI Taxonomy database. We agree that it can be a confusing description and we have refined it in the manuscript to improve clarity.

10. The authors mention that the parameters that they are able to asses are “limited to continuous variables (line 200)”. I wonder why they set such restrictions. Does it also mean that their system cannot handle nominal or categorical data?

At the moment, the correlation analysis only deals with the continuous variables. The nominal or categorical data are only used for visualization. We plan to add other analysis approaches, which use the categorical data in the future development.

11. I highly suggest the authors check QIIME2 visualization support since a lot of functionalities that they claim are already supported by QIIME2, for example the coloring of sample points according to the numerical or nominal parameters (line 238).

We agree that some visualization functions of MANTA can be also achieved by using QIIME2 as well as other tools such as R. However, MANTA’s main focus is the ability to data storage and data/analysis sharing among the project members. We have, nevertheless, added an appropriate discussion on these related tools in the revised manuscript.

12. The resolutions of the figures are very poor. Please consider making higher quality ones.

We did submit high quality images but the PDF processing steps may have decreased the image resolution. We will confirm the image quality when submitting the revised manuscript.

13. Due to poor figure quality and the inability to test the website, I cannot tell whether the authors provide significance metrics such as p-value/FDR or adjust R-squared. Please consider adding such metrics in order to help your users interpreting data if these are not supported.

Thank you for the suggestions. We have added p-value for the correlation in the application. We will also consider adding other significance metrics in the future developments.

14. The authors claim that “for users who wish to use the platform for analyzing their data, we also developed MANTA basic.” This is also very intriguing since it hinted that the existing MANTA website only supports the display of the 20-Japanese data, and that the users need to install their own system if they want to analyze their data. If this is true then I honestly do not know the existence meaning of the MANTA website. Please clarify if this is the case.

We apologize for the confusion. We explicitly named what the reviewer called the MANTA website “MDD” (MANTA demonstration database); MDD is designed for demonstrating the functionality of MANTA by using a small example data set. MANTA needs to be installed in a server setting or a local machine (MANTA basic) to be used for both storing and analyzing the data. We have presented a software program, instead of an online resource. We have extensively modified the manuscript to clarify these points.

15. I don’t understand what does “parameter grouping” mean during the comparison between MANTA and MANTA Basic. Please clarify.

To browse a large number of phenotypic parameters easily, MANTA implements a function of classifying them into groups (see Table 1). In MANTA basic, this function is unavailable due to the simplified data structure. We have added a footnote to comparison table (Table 2) to clarify this point.

16. (line 173) pageable  do you mean scrollable?

We agree that this term is ambiguous. We have used the word "scrollable" instead.

17. (line 192) 10 most abundant will  10 most abundant ones will

We have rephrased the sentence.

---

## [Decision Letter · Decision Letter 1]

16 Oct 2020

PONE-D-20-09945R1

MANTA, an integrative database and analysis platform that relates microbiome and phenotypic data

PLOS ONE

Dear Dr. Chen,

Thank you for submitting your manuscript to PLOS ONE. After careful consideration, we feel that it has merit but does not fully meet PLOS ONE’s publication criteria as it currently stands. Therefore, we invite you to submit a revised version of the manuscript that addresses the points raised during the review process.

We look forward to receiving your revised manuscript.

Kind regards,

Lingling An

Academic Editor

PLOS ONE

Reviewers' comments:

Reviewer's Responses to Questions

**Comments to the Author**

1. If the authors have adequately addressed your comments raised in a previous round of review and you feel that this manuscript is now acceptable for publication, you may indicate that here to bypass the “Comments to the Author” section, enter your conflict of interest statement in the “Confidential to Editor” section, and submit your "Accept" recommendation.

Reviewer #1: All comments have been addressed

Reviewer #2: (No Response)

Reviewer #3: (No Response)

2. Is the manuscript technically sound, and do the data support the conclusions?

Reviewer #1: Yes

Reviewer #2: Yes

Reviewer #3: Partly

3. Has the statistical analysis been performed appropriately and rigorously? 

Reviewer #1: I Don't Know

Reviewer #2: N/A

Reviewer #3: Yes

4. Have the authors made all data underlying the findings in their manuscript fully available?

Reviewer #1: Yes

Reviewer #2: Yes

Reviewer #3: No

5. Is the manuscript presented in an intelligible fashion and written in standard English?

Reviewer #1: Yes

Reviewer #2: Yes

Reviewer #3: Yes

6. Review Comments to the Author

Reviewer #1: I appreciate the changes authors have made in the revised MS. Thank you for clarifying my queries and restructuring the manuscript. If possible, the authors must give a statement indicating their plans on how long is the tool expected to be supported and maintained given the funding they have. This is important because many tools after publication are forgotten by developers. I hope that this tool will be maintained and updated for at least 5-10 years.

Reviewer #2: This revision has addressed most of my concerns. I can use the online manta website and the offline software. I suggest improving the documentation by at least provide:

1) How to generate the microbiota file (what software, what file format is accepted)?

2) What file format is expected for user’s phenotype file?

3) Explain “Parameter type”. What does “others” mean? How does MANTA treat “others” compared to other types? The example phenotypic file has 290 variables, and I suggest automatically inferring the parameter types in MANTA.

Reviewer #3: Upon this revision I now understand that the website described in the first version of the MANTA manuscript was totally fake--or I should say that it is a useless website only for demonstration purpose. The most critical point, as I identified, is that the installation procedure of the webpage components (jdk, tomcat, postgresql, data preprocessing, gwt compilation, to name just a few) can be much more challenging to users compared to ordinary pipelines. Due to this I do not know why users need to spend efforts to set up their own website--using command line pipelines can be easier than that. Thus, again, I doubt the very existence meaning of MANTA.

I have said it in my first review but allow me to say it again: please think and discuss with non-computational guys (or just let them try) in order to know their needs. Setting up a website can also be challenging even for people working on computational tasks. Please ponder the needs of non-computational people and design the system/pipeline accordingly.

7. PLOS authors have the option to publish the peer review history of their article (what does this mean?). If published, this will include your full peer review and any attached files.

Reviewer #1: No

Reviewer #2: No

Reviewer #3: No

---

## [Author Response · Author response to Decision Letter 1]

26 Oct 2020

Response to Reviewers

Details:

Reviewer #1: I appreciate the changes authors have made in the revised MS. Thank you for clarifying my queries and restructuring the manuscript. If possible, the authors must give a statement indicating their plans on how long is the tool expected to be supported and maintained given the funding they have. This is important because many tools after publication are forgotten by developers. I hope that this tool will be maintained and updated for at least 5-10 years.

Our current funding supports tool development and related activities for the next five years. We hope that we can continuously improve and maintain the MANTA project for at least 5-10 years. 

Reviewer #2: This revision has addressed most of my concerns. I can use the online manta website and the offline software. I suggest improving the documentation by at least provide:

1) How to generate the microbiota file (what software, what file format is accepted)?

2) What file format is expected for user’s phenotype file?

3) Explain “Parameter type”. What does “others” mean? How does MANTA treat “others” compared to other types? The example phenotypic file has 290 variables, and I suggest automatically inferring the parameter types in MANTA.

Thank you for the suggestions. We have improved the online documentation to address those issues. The following explanation has been added: 

1) The currently accepted microbiota file format is a tab-delimited table format (see the sample data as an example), which can be easily converted from the BIOM (The Biological Observation Matrix) format file (https://biom-format.org/). The BIOM format is adopted by several popular projects including QIIME and MG-RAST. Kraken/Bracken produces results in a different format and we are working on enabling the system to import this type of data.

2) At the moment, the accepted format is a tab delimited table format that contains different parameters in each column and different samples in each row.

3) The “others” type was designed for the data types that do not belong to any of the rest. At the moment, this type of data is treated in the same way as the type ‘free text’, and not used for any correlation calculation.

Regarding the auto assignment of the data types, we will add this feature in a future release.

Reviewer #3: Upon this revision I now understand that the website described in the first version of the MANTA manuscript was totally fake--or I should say that it is a useless website only for demonstration purpose. The most critical point, as I identified, is that the installation procedure of the webpage components (jdk, tomcat, postgresql, data preprocessing, gwt compilation, to name just a few) can be much more challenging to users compared to ordinary pipelines. Due to this I do not know why users need to spend efforts to set up their own website--using command line pipelines can be easier than that. Thus, again, I doubt the very existence meaning of MANTA.

I have said it in my first review but allow me to say it again: please think and discuss with non-computational guys (or just let them try) in order to know their needs. Setting up a website can also be challenging even for people working on computational tasks. Please ponder the needs of non-computational people and design the system/pipeline accordingly.

Our demonstration database includes real-life (albeit small sample size) data. We believe that such a website, in line with many other open source projects, will be useful for showing key features of the software tool. We appreciate the hurdles for the non-computational users. In the revised version of the manuscript, we already discussed the importance of addressing the needs of non-technical users (in the paragraph starting from line 371), and this is exactly why we developed and released a standalone version named MANTA basic; the user can simply download a single all-in-one package and launch the application with a double click. To clarify these points, we have updated the documentation on our website.

---

## [Decision Letter · Decision Letter 2]

25 Nov 2020

MANTA, an integrative database and analysis platform that relates microbiome and phenotypic data

PONE-D-20-09945R2

Dear Dr. Chen,

We’re pleased to inform you that your manuscript has been judged scientifically suitable for publication and will be formally accepted for publication once it meets all outstanding technical requirements.

Kind regards,

Lingling An

Academic Editor

PLOS ONE

Additional Editor Comments (optional):

Reviewers' comments:

Reviewer's Responses to Questions

**Comments to the Author**

1. If the authors have adequately addressed your comments raised in a previous round of review and you feel that this manuscript is now acceptable for publication, you may indicate that here to bypass the “Comments to the Author” section, enter your conflict of interest statement in the “Confidential to Editor” section, and submit your "Accept" recommendation.

Reviewer #2: All comments have been addressed

2. Is the manuscript technically sound, and do the data support the conclusions?

Reviewer #2: Yes

3. Has the statistical analysis been performed appropriately and rigorously? 

Reviewer #2: N/A

4. Have the authors made all data underlying the findings in their manuscript fully available?

Reviewer #2: Yes

5. Is the manuscript presented in an intelligible fashion and written in standard English?

Reviewer #2: Yes

6. Review Comments to the Author

Reviewer #2: This revision has addressed all my concerns. In the future, I hope that the authors can continue maintaining the website and add more microbiome projects in the MANTA database.

7. PLOS authors have the option to publish the peer review history of their article (what does this mean?). If published, this will include your full peer review and any attached files.

Reviewer #2: No

---

## [Editor Report · Acceptance letter]

27 Nov 2020

PONE-D-20-09945R2 

MANTA, an integrative database and analysis platform that relates microbiome and phenotypic data 

Dear Dr. Chen:

I'm pleased to inform you that your manuscript has been deemed suitable for publication in PLOS ONE. Congratulations! Your manuscript is now with our production department. 

Kind regards, 

on behalf of

Dr. Lingling An 

Academic Editor

PLOS ONE